# Offline Change Detection under Contamination

**Sujay Bhatt, Guanhua Fang, Ping Li**

Cognitive Computing Lab
Baidu Research
10900 NE 8th St. Bellevue, WA 98004, USA
{sujaybhatt.hr, fanggh2018, pingli98}@gmail.com

## Abstract

We propose a non-parametric and robust change detection algorithm to detect multiple change points in time series data under contamination. The contamination model is sufficiently general, in that, the most common model used in the context of change detection – Huber contamination model – is a special case. Also, the contamination model is oblivious and arbitrary. The change detection algorithm is designed for the offline setting, where the objective is to detect changes when all data are received. We only make weak moment assumptions on the inliers (uncorrupted data) to handle a large class of distributions. The robust scan statistic in the algorithm is fashioned using mean estimators based on influence functions. We establish the consistency of the estimated change point indexes as the number of samples increases, and provide empirical evidence to support the consistency results.

## 1 INTRODUCTION

Change point detection in time series data is the task of identifying changes in the underlying data generation model and can be traced back to the initial work of Page [1954, 1955] in the context of statistical process/quality control. This simple and elegant framework has been deployed in diverse applications such as bioinformatics [Vert and Bleakley, 2010], finance [Pástor and Stambaugh, 2001, Pepelyshev and Polunchenko, 2017], biology [Siegmund, 2013], climatology [Verbesselt et al., 2010], metric learning [Lajugie et al., 2014]; to name a few.

Change detection methods are mainly classified into online and offline settings. In the online setting, the aim is to detect changes as soon as they occur in real-time by optimizing an objective that trades-off detection delay and false alarm; see Poor and Hadjiliadis [2008] for a detailed introduction and Xie et al. [2021] for a survey of recent developments. In contrast, in the offline setting, the changes need to be detected in a retrospective manner by 'segmenting' the entire dataset [Aminikhanghahi and Cook, 2017]. Here the objective is to design consistent algorithms and empirically validate using well-known metrics such as F1-Score [de Bem et al., 2020], Hausdorff metric [Harchaoui and Lévy-Leduc, 2010], etc; see Truong et al. [2020] for detailed overview of the methods and recent developments.

In this work, we consider the offline setting and contribute to the literature by relaxing the common assumptions. To motivate the setup considered with an example, consider monitoring mean shifts in non-stationary processes using Wireless Sensor Networks (WSN) [Akyildiz and Vuran, 2010, Cui and Xie, 2019]. In addition to the inherent challenges such as dealing with non-i.i.d data [Tartakovsky, 2019] and heavy-tails [Fearnhead and Rigaill, 2019, Bhatt et al., 2021], modern machine learning applications have to deal with the introduction of adversarial examples in the dataset [Kurakin et al., 2017, Jia and Liang, 2017]. Specifically, when the WSNs are used in applications such as healthcare (EEG/ fMRI), environmental impact monitoring, energy consumption, etc; the sensors are typically deployed in harsh conditions. This increases data corruption or erroneous readings during transmission. When the data from a collection of near-by sensors are logged for surveillance and event-classification, any inference procedure should account for the following salient features: *non-i.i.d, outliers, and adversarial contamination*. This motivates the development of change detection algorithms in the offline setting [Aminikhanghahi and Cook, 2017, Truong et al., 2020] that can tackle all the above challenges in a systematic manner.

### 1.1 MAIN RESULTS

We propose a non-parametric change detection algorithm that can deal with non-i.i.d data, outliers, and a weak form of adversarial contamination to identify the change points in a consistent manner. Specifically, our contributions are:

*Accepted for the 38th Conference on Uncertainty in Artificial Intelligence* (UAI 2022).

1. Non-parametric algorithms feature a key quantity known as *scan statistic*, for example CUSUM statistic of Page [1954], which is required to 'scan' the dataset to identify the change points. We propose a scan statistic based on influence functions proposed by Catoni [2012] that can handle outliers and heavy-tails, to deal additionally with contamination. We consider a contamination model, where the outliers (corrupted data) are correlated to each other and to inliers (uncorrupted data). The inliers can also be correlated to one another. The resulting robust non-parametric algorithm `RC-Cat` announces a change if the scan statistic exceeds a pre-specified threshold, provided the scan statistic is a local maximum. This additional sophistication of local search methods was the introduced in Niu and Zhang [2012] and developed for the robust version in Li and Yu [2021], to mainly avoid overestimation of change points.

2. A natural way to theoretically evaluate change detection algorithms is to establish consistency of the estimated change point indexes as the number of samples increases. In particular, we show that `RC-Cat` is consistent in the presence of contamination, i.e, as the number of data points $n \uparrow \infty$,

$$\mathbb{P}\Big(\widehat{K} = K, \max_{k=1}^{\widehat{K}} |\widehat{\tau}_k - \tau_k| \leq w\Big) \to 1,$$

where $K$ is the number of true change points located at $\tau_k$, $k \in \{1, \cdots, K\}$ and $\widehat{K}$ is the number of detected change points announced at $\widehat{\tau}_k$, $k \in \{1, \cdots, \widehat{K}\}$, and $w > 0$ is parameter that related to the window length considered.

## 1.2 RELATED LITERATURE

In the context of robust change detection, a common model of contamination that is considered to design algorithm is the Huber contamination model [Huber, 1964]. In this model, the data generation model is a mixture model $(1-\eta)F + \eta Q$, where $F$ is the true distribution before the change and $Q$ is any arbitrary distribution with a probability $\eta$. Using such a model of contamination, Hušková [2013] make use of M-estimation idea from robust statistics [Huber, 2004] to address the change point detection problem in the context of regression. Fearnhead and Rigaill [2019] consider penalized M-estimation based procedure that can deal with outliers. [Prasad et al., 2020] showed that Huber contamination model is equivalent to assuming a heavy-tailed noise for the i.i.d data. In light of this, Yu and Chen [2022] propose a scan statistic based on U-statistics to deal with heavy-tailed noise distributions. The setup and analysis considered in this work is closest to Li and Yu [2021], however, with the following key differences:

- Li and Yu [2021] consider change detection under i.i.d data. While this is a useful starting point, it only serves as a crude approximation when the data is gathered from heterogeneous sources [Mercier et al., 2008] and is in general non-i.i.d [Tartakovsky, 2019].

- The scan statistics is fashioned using the robust estimator (RUME) in Prasad et al. [2020]. RUME uses half of the samples to identify the shortest interval containing at least $(1 - \eta)n$ fraction of the points, and then the remaining half of the points is used to return an estimate of the mean. While this is acceptable in the case of robust mean estimation, it has clear disadvantages in the context of change point detection, where the initial segregation might hide/ remove the true change points. Another feature of RUME is that the amount of contamination that the estimator can handle is limited, and this limits the applicability in many applications.

In contrast, our algorithm deals with non-i.i.d inliers and contamination, where the inliers only have a bounded second moment. Also, unlike Li and Yu [2021], we do not segregate the data for robust mean estimation, which avoids the problem of loosing change points. Using empirical results, we further show that, not only the proposed algorithm is more general than that in Li and Yu [2021], it is faster and obtains better detection performance across different settings.

## 2 MEAN ESTIMATION UNDER CONTAMINATION

In this section, we propose a robust mean estimator that can deal with **non-i.i.d data** with arbitrary contamination. The estimator is based on influence functions proposed in Catoni [2012] and Catoni and Giulini [2017], and is further developed in Bhatt et al. [2022a,b]. Let $\{X_t\}_{t=1}^n$ be a collection of real-valued random variables. Let $\mathcal{F}_0$ denote the trivial sigma algebra, and let $\mathcal{F}_t$ denote the sigma-algebra generated by the set $\{X_1, X_2, \cdots, X_t\}$, whence $X_t$ is $\mathcal{F}_t$−measurable. Let $[n] := \{1, 2, \cdots, n\}$.

**C1.** The set $\{X_t\}_{t\in[n]}$ is such that the (unknown) conditional expected value

$$\forall\, t \in [n], \;\; \mathbb{E}\Big[X_t | \mathcal{F}_{t-1}\Big] = \mu_t.$$

**C2.** The conditional second moment of $X_t$ is bounded, i.e, for a known $\mathcal{M} > 0$,

$$\forall\, t \in [n], \;\; \mathbb{E}\Big[X_t^2 | \mathcal{F}_{t-1}\Big] \leq \mathcal{M}.$$

It is easy to see that i.i.d is a special case that satisfies **C1** and **C2**. However, the model allows for more general dependencies, see Seldin et al. [2012].

## 2.1 CONTAMINATION MODEL

We assume that for some corruption rate $0 < \eta < 1$, an adversary may change at most $\eta k$ of any sub-sequence of

$\{X_t\}_{t\in[n]}$ with length $k \geq k_0$, to arbitrary values. The resulting set of observations will be $\widetilde{X}_1, \widetilde{X}_2, \cdots, \widetilde{X}_n$, so that

$$\sup_{i\in[n-k]} \sum_{j=1}^{k} \mathcal{I}\left(\widetilde{X}_{i+j} \neq X_{i+j}\right) \leq \eta k, \qquad (1)$$

where $\mathcal{I}(\cdot)$ denotes the indicator function and $k \geq k_0$ with $k_0$ being a fixed integer such that $k_0 = \Omega(\log n)$. The task is to estimate the true mean $\mu := \frac{1}{n}\sum_{i=1}^{n}\mu_i$ based on the observations $\widetilde{X}_1, \widetilde{X}_2, \cdots, \widetilde{X}_n$. This contamination model is widely studied in machine learning for i.i.d data [Charikar et al., 2017, Hopkins and Li, 2018, Lugosi and Mendelson, 2021]. This model is similar to $\mathcal{I} \cup \mathcal{O}$ model of Lecué and Lerasle [2019] and also shares similarities with the Huber contamination model [Huber, 1964]. While the contamination can be arbitrary, we do not allow the possibility where the adversary corrupts a fraction of the sample possibly with the knowledge of the whole dataset to intentionally hide the change points, i.e, the contamination is *weakly adversarial*.

**Remark**. The well-known Huber contamination model in change detection [Li and Yu, 2021] is a special case of the considered adversarial model. Let $\epsilon \in [0, 1]$ denote the outlier distribution probability in the Huber contamination model, i.e, the data is generated as $(1-\epsilon)P+\epsilon Q$, where $P$ is the true distribution and $Q$ is any arbitrary distribution. Let the empirical fraction $\widehat{\epsilon}_n = \sup_{i\in[n-k]} \frac{1}{k}\sum_{j=1}^{k}\mathcal{I}(\widetilde{X}_{i+j} \neq X_{i+j})$. According to a recent result in Bhatt et al. [2022b], with probability at least $1 - \beta$, we have for all $k$

$$\widehat{\epsilon}_n \leq \epsilon + \underbrace{1.7\sqrt{\epsilon(1-\epsilon)}\sqrt{\frac{\log(\log(2n)) + 0.72\log\frac{10.4n}{\beta}}{k}}}_{:=f(\beta,\epsilon,k)}.$$

Fix $k_0 > c\log(n/\beta)$ and set

$$\epsilon + f(\beta, \epsilon, k_0) =: \eta.$$

Clearly, for all $k \geq k_0$, we have the corruption fraction $\widehat{\epsilon}_n$ to be at most $\eta$ with a very high probability.

## 2.2 MEAN ESTIMATION WITH INFLUENCE FUNCTIONS

The idea of using influence functions for robust mean estimation is not new [Huber, 2004]. However these M-estimators are unable to scale gracefully with dimension [Maronna, 1976, Donoho and Gasko, 1992], and Prasad et al. [2020] show that the bias scales polynomially with dimension. This led to the development of a class of M-estimators introduced by Catoni [2012] that can be used to obtain dimension-free bounds in the vector settings [Catoni and Giulini, 2017]. With a similar future objective in mind, we make use of the influence functions proposed in these works to fashion a robust estimator that has minimax optimal asymptotic bias

in the contamination parameter when the data sequence is more general than i.i.d.

Consider a non-decreasing function $\psi : \mathbb{R} \to \mathbb{R}$ such that

$$-\log(1 - x + x^2/2) \leq \psi(x) \leq \log(1 + x + x^2/2)$$

for all $x \in \mathbb{R}$ as in Catoni [2012]. One can choose such a function that is bounded: specifically, we assume that for some $0 < A < \infty$,

$$|\psi(x)| \leq A \text{ for all } x \in \mathbb{R}. \qquad (2)$$

From Catoni [2012], the narrowest possible choice for the influence function has $A = \log 2$, and is given by

$$\psi(x) = \begin{cases} -\log(1 - x + x^2/2), & 0 \leq x \leq 1, \\ \log(2), & x \geq 1, \\ -\psi(-x), & -1 \leq x \leq 0, \\ -\log(2), & x \leq -1. \end{cases} \qquad (3)$$

We consider an estimator based on soft-truncation after re-scaling, defined by

$$\widehat{\mu}_\eta := \frac{\alpha}{n}\sum_{i=1}^{n}\psi(\frac{\widetilde{X}_i}{\alpha}), \qquad (4)$$

where $\alpha > 0$ is a re-scaling parameter, and the uncontaminated version is given as

$$\widehat{\mu} := \frac{\alpha}{n}\sum_{i=1}^{n}\psi(\frac{X_i}{\alpha}). \qquad (5)$$

In the absence of contamination, depending on the choice of $\psi(\cdot)$ and $\alpha$, the estimator (5) can closely approximate the empirical mean; see Holland [2019] for example. Similar estimator for i.i.d data was considered in Holland [2019], where the deviation bounds were characterized using well-known PAC Bayesian inequalities inspired by Donsker-Varadhan's variational formula [Catoni, 2004, Dupuis and Ellis, 2011]. However, since the data are not i.i.d in our case, we need a different approach to characterize the deviations, and this is the main contribution of this section.

**Theorem 1.** *Consider a collection of random variables* $\{\widetilde{X}_t\}_{t\in[n]}$. *Let* $\alpha = \sqrt{\frac{\mathcal{M}}{2\left(\frac{\log(2/\delta)}{n}+2A\eta\right)}}$ *and* $\delta \in (0, 1)$. *The estimator* (4) *satisfies*

$$|\widehat{\mu}_\eta - \mu| \leq \sqrt{2\mathcal{M}\left(\frac{\log(2/\delta)}{n} + 2A\eta\right)}, \qquad (6)$$

*with probability at least* $1 - \delta$.

A high-probability deviation bound for $\widehat{\mu}$, i.e, in the absence of contamination is first characterized as a function of $\alpha$, whence we obtain

$$|\widehat{\mu} - \mu| \leq \frac{\mathcal{M}}{2\alpha} + \frac{\alpha\log(2/\delta)}{n}.$$

From (2), we have the following relation

$$|\widehat{\mu}_\eta - \mu| \le |\widehat{\mu} - \mu| + 2A\eta\alpha.$$

This provides the deviation bound of the overall soft-truncation estimator (4).

**Corollary 2.** *Let $B > 1$. Under the assumptions as in Theorem 1 such that (6) holds, we have with probability at least $1 - 2\exp\left(-\frac{A\eta}{B}n\right)$*

$$|\widehat{\mu}_\eta - \mu| \le c_0\sqrt{\mathcal{M}\eta}, \tag{7}$$

*where $c_0^2 := 2A(1/B + 2)$.*

Corollary 2 obtains the deviation purely in terms of the contamination fraction, and will be useful later in establishing the consistency of change detection algorithms. Another useful feature is that it informs the choice of segmentation window length that guarantees a tight deviation characterization.

From (6), it is clear that there is an asymptotic ($n \uparrow \infty$) bias of $O(\sqrt{\mathcal{M}\eta})$ associated with the estimator owing to contamination. Also, when $\mu_t \in [0, 1]$ with a possibly heavy tail martingale difference noise– a common assumption in bandits [Lattimore and Szepesvári, 2020] and reinforcement learning [Agarwal et al., 2019]– the deviation bound and hence the bias can be written in terms of the (conditional) variance $\sigma^2$ as $O(\sigma\sqrt{\eta})$ by using the standard $C_r$ inequality[1]. This matches the minimax lower bound [Diakonikolas et al., 2017, Hopkins and Li, 2018] that is shown to be information theoretically optimal.

However, in general, as the deviation (6) depends on the non-centered moment, it is sensitive to the location of the distribution. Catoni and Giulini [2017] propose a 'shifting-device' approach to obtain centered estimates that can be used to obtain a deviation bound in terms of the conditional variance. This has been used for PAC-Bayesian analysis using influence functions in Holland [2019].

**Theorem 3.** *Consider the set of r.vs $\{\widetilde{X}_t\}_{t\in[n]}$ such that $\mu_t = \mu$, $\forall$ $t$. Additionally, let $\mathcal{V}$ denote an upper bound on conditional variance of the uncontaminated random variables. Let $0 < k < n$ denote the length of the data to create a shifting device. Let $\alpha = \sqrt{\frac{(\mathcal{V}+\vartheta_k^2)}{2\left(\frac{\log(2/\delta)}{n-k}+2A\eta\right)}}$ with $\vartheta_k = \sqrt{2\mathcal{M}\left(\frac{\log(2/\delta)}{k}+2A\eta\right)}$. The estimator (4) satisfies*

$$|\widehat{\mu}_\eta - \mu| \le \sqrt{2(\mathcal{V}+\vartheta_k^2)\left(\frac{\log(4/\delta)}{n-k}+2A\eta\right)}, \tag{8}$$

*with probability at least $1 - \delta$.*

---

[1]For any random variable $Y$, real number $\gamma$, and $r > 0$,

$$|Y|^r \le \max\{2^{r-1}, 1\}(|Y - \gamma|^r + |\gamma|^r).$$

Theorem 3 provides a deviation bound as a function of the conditional variance. When the contamination level $\eta$ is negligible, a judicious choice of $k$ will lessen the dependence on the raw moments, and the conditional variance in the deviation term. Theorem 3 works to combat sensitivity to the distribution location. A procedure to obtain an estimator having the deviation bound as in (8) is given as follows:

i.) Shifting-Device: Let $\{\widetilde{X}_i\}_{i=1}^k$ denote a sub-set of the collection. Compute a soft-truncated estimate using these $k$ samples,

$$\bar{\mu}_\eta := \frac{\bar{\alpha}}{k}\sum_{i=1}^{k}\psi\left(\frac{\widetilde{X}_i}{\bar{\alpha}}\right),$$

where $\bar{\alpha}$ is informed by Theorem 1.

ii.) Shift the remaining $n - k$ samples by $\bar{\mu}_\eta$, i.e, $\widetilde{X}_i' = \widetilde{X}_i - \bar{\mu}_\eta$, whence the conditional second moment of this data is now bounded by $(\mathcal{V} + \vartheta_k^2)$. Computing the soft-truncated estimate of this data

$$\mu_\eta' := \frac{\alpha'}{n-k}\sum_{i=k+1}^{n}\psi\left(\frac{\widetilde{X}_i'}{\alpha'}\right),$$

where $\alpha'$ is informed by Theorem 3.

iii.) Estimator $\widehat{\mu}_\eta = \mu_\eta' + \bar{\mu}_\eta$ has the desired properties.

## 3 OFFLINE CHANGE DETECTION

In the rest of the paper, we assume that the contamination model used by the adversary is as in (1). We first provide an algorithm based on the robust estimation techniques discussed in Section 2.2, and later establish the theoretical properties of the algorithm.

### 3.1 THE PROPOSED ALGORITHM

Algorithm 1 is an offline robust change detection algorithm that can handle $\eta$ fraction of weakly adversarial contamination when the data is not necessarily i.i.d. The methodology is inspired by Niu and Zhang [2012] and Li and Yu [2021], which handle the uncontaminated and weak contamination situations respectively.

Algorithm 1 is an intuitive solution that combines local and global search methods in a non-parametric manner to identify the change points. It works as follows: The dataset is scanned using the scan statistic $S_w(\cdot)$, which is the absolute difference between the robust estimates of mean over specified length $w$. Here the estimator over length $w > 0$,

$$\Psi(\{\widetilde{X}_i\}_{i=1}^w) := \frac{\alpha}{w}\sum_{i=1}^{w}\psi\left(\frac{\widetilde{X}_i}{\alpha}\right),$$

with one possible choice of $\psi(\cdot)$ given by (3). The nature of the (non-parametric) scan statistic, where normalized

**Algorithm 1** Robust Change Detection with Catoni (`RC-Cat`)

---

1: **Input:** $\{\widetilde{X}\}_{i=1}^n$, $b$(threshold) $> 0$, $2w$(window) $> 0$, $\eta \in (0,1), \lambda \geq 1$
2: $\mathcal{L} \leftarrow \emptyset, \mathcal{G} \leftarrow \emptyset$
3: **for** $j \in \{w+1, \cdots, n-w\}$ **do**
4: $\quad S_w(j) \leftarrow \left| \Psi\left(\{\widetilde{X}_i\}_{i=j+1}^{j+w}\right) - \Psi\left(\{\widetilde{X}_i\}_{i=j-w}^{j-1}\right) \right|$
5: **end for**
6: **for** $j \in \{\lambda w+1, \cdots, n-\lambda w\}$ **do**
7: $\quad$ **if** $j$ is a $\lambda w-$local maximizer of $S_w(j)$ **then**
8: $\quad\quad \mathcal{L} \leftarrow \mathcal{L} \cup \{j\}$
9: $\quad$ **end if**
10: **end for**
11: **for** $k \in \mathcal{L}$ **do**
12: $\quad$ **if** $S_w(k) > b$ **then**
13: $\quad\quad \mathcal{G} \leftarrow \mathcal{G} \cup \{k\}$
14: $\quad$ **end if**
15: **end for**
16: **Output:** $\mathcal{G}$

---

estimates of equal length of samples are compared, is well-studied in the literature. For example, Cao et al. [2019] make use of similar ideas for empirical means of independent sub-gaussian distributions to detect changes in the mean in multi-armed bandit problems, while Niu and Zhang [2012] consider an application in bioinformatics. The robust scan statistic is closest to that in Li and Yu [2021], except with a few key differences: (i) There is no sample splitting to estimate the location parameter using RUME [Prasad et al., 2020]. In Li and Yu [2021], the data over $w/2$ is used to simply identify a high-confidence interval, and the remaining $w/2$ portion is used to calculate the robust mean. This not only increases the variance of the estimator, but also may hide/ remove change points depending on which of $w/2$ points is selected. This affects the detection delay and hence the consistency. (ii) The worst case computational complexity of `RC-Cat` is $O(nw)$, whereas the worst case complexity in case of Li and Yu [2021] is $O(n^2 w \log(w))$. Here the $w \log(w)$ is from ranking the data to find the shortest interval involved in RUME. Note that the state-of-the-art, such as penalized bi-weight loss, methods have a computational complexity of $O(n^3)$ [Fearnhead and Rigaill, 2019].

The $\lambda w-$local maximizer is inspired by Niu and Zhang [2012] and also appears in Li and Yu [2021]. $S_w(j)$ is a $h-$local maximizer if $S_w(j) \geq S_w(k)$ for all $k \in (j-h, j+h)$, and is motivated by two key ideas: (i) It helps to avoid overestimating the number of change points. (ii) It helps to localize change points with a high probability.

## 3.2 THE ANALYSIS

`RC-Cat` is a computationally appealing solution to offline change detection. In this section, we establish that it is con-

sistent as well, i.e, as the number of data samples increase, the regime changes are identified within a prescribed margin with a high probability. We need to make a few standard assumptions to enable this result.

Let $K$ be the number of true change points located at $\tau_k$, $k \in \{1, \cdots, K\}$ with $\tau_0 = 0$ and $\tau_{K+1} = n$. Let $\widehat{K}$ be the number of detected change points announced at $\widehat{\tau}_k$, $k \in \{1, \cdots, \widehat{K}\}$ by `RC-Cat`. Let the collections be denoted as $\mathcal{K} := \{\tau_1, \cdots, \tau_K\}$ and $\widehat{\mathcal{K}} := \{\widehat{\tau}_1, \widehat{\tau}_2, \cdots, \widehat{\tau}_{\widehat{K}}\}$ respectively. Let the minimal spacing be denoted as $\delta = \min_{k \in \mathcal{K}} |\tau_{k+1} - \tau_k|$ and the jump size be denoted as $\theta = \min_{k \in \mathcal{K}} |\mu_{\tau_k+1} - \mu_{\tau_k}|$.

**A1.** The conditional expectation in **C1** is constant between change points, that is,
for $k \in \{1, 2, \cdots, K+1\}$,

$$\mu_t = \mu_{\tau_{k-1}}, \ \ \forall \ \tau_{k-1} \leq t < \tau_k.$$

**A2.** The minimal spacing $\delta > \lambda w$ for some $\lambda \geq 2$.

**A3.** The jump size $\theta > \sqrt{3b}$, where $b$ is the threshold.

**A2** essentially says that the process has slow changes and **A3** is related to detectability. The assumptions **A2** and **A3** are intuitive and standard in the change detection literature [Niu and Zhang, 2012, Cao et al., 2019, Yu, 2020, Li and Yu, 2021], while **A1** simplifies exposition. While these conditions are necessary for characterizing the theoretical properties, deviations from these assumptions do not drastically affect the empirical performance. Also, we should mention that, **A1** can be relaxed to allow small perturbations between change points for the conditional expectation, and the same analysis carries over.

**Theorem 4.** *Let $\{\widetilde{X}_i\}_{i \in [n]}$ be the collection of r.vs input to* `RC-Cat`. *Let the threshold $b = 2c_0\sqrt{\mathcal{M}\eta}$ and window $w \geq c_1 \log(n)/\eta$. Under assumptions A1 - A3, it holds that*

$$\mathbb{P}\left(\widehat{K} = K, \max_{k=1}^{\widehat{K}} |\widehat{\tau}_k - \tau_k| \leq w\right) \geq 1 - n^{-c_1}. \quad (9)$$

Theorem 4 shows that for large dataset, `RC-Cat` identifies the change points or the segments to within specified tolerance with a high probability. In other words, as $n \uparrow \infty$,

$$\mathbb{P}\left(\widehat{K} = K, \max_{k=1}^{\widehat{K}} |\widehat{\tau}_k - \tau_k| \leq w\right) \to 1.$$

Due to the nature of the robust estimator used in the scan statistic, `RC-Cat` can handle data from heterogeneous sources and non-i.i.d as specified by **C1** and **C2**.

**Corollary 5.** *Let $\mathcal{K} = \emptyset$. Under the same assumptions as in Theorem 4, there is a constant $c_1 > 0$ such that*

$$\mathbb{P}(\widehat{K} = 0) \geq 1 - n^{-c_1}.$$

Corollary 5 says that when $\mathcal{K} = \emptyset$, the proposed algorithm `RC-Cat` is still consistent, and provides an upper bound on the false detection probability.

# 4 PROOFS OF MAIN RESULTS

In this section, we provide the proofs of the main results. The proof of Theorem 1 builds on the standard martingale analysis [Freedman, 1975, Seldin et al., 2012] to establish the bounds for bounded functions of real-valued random variables. The key idea is to make use of the fact that the influence function $\psi(\cdot)$ is bounded by logarithmic functions, and to construct a supermartingale as a function of $\psi(\cdot)$. The result then follows using Markov's inequality. The proof of Theorem 4 closely follows Niu and Zhang [2012], but under weaker assumptions on the data and the parameters. Also, in comparison with Li and Yu [2021], for a fixed confidence $\delta$, RC-Cat achieves consistency even over smaller sized datasets.

## 4.1 PROOF OF THEOREM 1

Before establishing the result, we will first characterize the high-probability deviation bound for the robust estimator in the absence of contamination as a function of $\alpha$. This is given as Lemma 6.

**Lemma 6.** *Let the set of r.vs $\{X_t\}_{t \in [n]}$ satisfy **C1** and **C2**. For $\alpha > 0$ and $\delta \in (0,1)$, the estimator (5) satisfies with probability at least $1 - \delta$*

$$|\widehat{\mu} - \mu| \leq \frac{\mathcal{M}}{2\alpha} + \frac{\alpha \log(2/\delta)}{n}.$$

*Proof.* For any $t \leq n$, we have the following using the upper bound on the influence function $\psi(\cdot)$,

$$\mathbb{E}\Big[\exp\Big(\psi\Big(\frac{X_t}{\alpha}\Big)\Big)\Big|\mathcal{F}_{t-1}\Big] \leq \mathbb{E}\Big[1 + \frac{X_t}{\alpha} + \frac{X_t^2}{2\alpha^2}\Big|\mathcal{F}_{t-1}\Big],$$
$$\leq 1 + \frac{\mu_t}{\alpha} + \frac{1}{2\alpha^2}\mathbb{E}\Big[X_t^2\Big|\mathcal{F}_{t-1}\Big].$$

Using the fact that $1 + x \leq e^x$ for all $x \in \mathbb{R}$, we have using **C2**

$$\mathbb{E}\Big[\exp\Big(\psi\Big(\frac{X_t}{\alpha}\Big)\Big)\Big|\mathcal{F}_{t-1}\Big] \leq \exp\Big(\frac{\mu_t}{\alpha} + \frac{\mathcal{M}}{2\alpha^2}\Big). \quad (10)$$

Construct a sequence of random variables $Y_t$ as follows: $Y_0 = 1$ and for $t \geq 1$,

$$Y_t = Y_{t-1} \exp\Big(\psi\Big(\frac{X_t}{\alpha}\Big)\Big) \exp\Big(-\Big(\frac{\mu_t}{\alpha} + \frac{\mathcal{M}}{2\alpha^2}\Big)\Big).$$

Clearly, $\mathbb{E}\Big[Y_t\Big|\mathcal{F}_{t-1}\Big] \leq Y_{t-1}$ as $Y_{t-1}$ is $\mathcal{F}_{t-1}$ measurable and (10) holds. We have that the unconditional expectation

$$\mathbb{E}[Y_n] \leq \mathbb{E}[Y_1] \leq \cdots \leq \mathbb{E}[Y_0] = 1.$$

Recursively, $Y_n$ is expressed as

$$Y_n = \exp\Big(\sum_{t=1}^n \psi\Big(\frac{X_t}{\alpha}\Big)\Big) \exp\Big(-\Big(\frac{n\mu}{\alpha} + \frac{n\mathcal{M}}{2\alpha^2}\Big)\Big),$$
$$= \exp\Big(\frac{n\widehat{\mu}}{\alpha}\Big) \exp\Big(-\Big(\frac{n\mu}{\alpha} + \frac{n\mathcal{M}}{2\alpha^2}\Big)\Big).$$

Here $\widehat{\mu}$ is given as in (5) and $\mu = \frac{1}{n}\sum_{t=1}^n \mu_t$. By Markov's inequality, we have that

$$\mathbb{P}(Y_n \geq 2/\delta) \leq \frac{\delta \mathbb{E}[Y_n]}{2} \leq \frac{\delta}{2}.$$

In other words, we have that

$$\mathbb{P}\Big(\frac{n\widehat{\mu}}{\alpha} \geq \frac{n\mu}{\alpha} + \frac{n\mathcal{M}}{2\alpha^2} + \log(2/\delta)\Big) \leq \frac{\delta}{2}.$$

Dividing by $\frac{n}{\alpha}$ gives the deviation in one direction. Using the lower bound on the influence function, we have that

$$\mathbb{E}\Big[\exp\Big(-\psi\Big(\frac{X_t}{\alpha}\Big)\Big)\Big|\mathcal{F}_{t-1}\Big] \leq \exp\Big(-\frac{\mu_t}{\alpha} + \frac{\mathcal{M}}{2\alpha^2}\Big).$$

Analogous arguments establish the deviation of the estimator $\widehat{\mu}$ in the other direction, whence

$$\mathbb{P}\Big(\frac{n\mu}{\alpha} - \frac{n\widehat{\mu}}{\alpha} \geq \frac{n\mathcal{M}}{2\alpha^2} + \log(2/\delta)\Big) \leq \frac{\delta}{2}.$$

The result follows. $\qquad\square$

From (2), we have the following relation

$$|\widehat{\mu}_\eta - \mu| \leq |\widehat{\mu} - \mu| + 2A\eta\alpha. \quad (11)$$

From (6) and (11), we have with probability at least $1 - \delta$,

$$|\widehat{\mu}_\eta - \mu| \leq \frac{\mathcal{M}}{2\alpha} + \frac{\alpha \log(2/\delta)}{n} + 2A\eta\alpha.$$

The main result holds by setting $\alpha$.

## 4.2 PROOF OF COROLLARY 2

From Theorem 1, we have with probability at least $1 - \delta$

$$|\widehat{\mu}_\eta - \mu| \leq \sqrt{2\mathcal{M}\Big(\frac{\log(2/\delta)}{n} + 2A\eta\Big)}.$$

By choosing $n \geq \frac{B}{A\eta} \log(2/\delta)$, we have the result.

## 4.3 PROOF OF THEOREM 4

The proof is established using the following reasoning. Let $T := \{x : |x - \tau_k| > w, \forall k \in \mathcal{K}\}$ denote the set of all points that are at least $w-$away from the true change points. Consider the following events,

$$\mathcal{E}_1(x) = \{S_w(x) < b\},$$
$$\mathcal{E}_2(y) = \{S_w(y) > b\},$$
$$\mathcal{E}_n = \Big(\cap_{k=1}^K \mathcal{E}_2(\tau_k)\Big) \cap \Big(\cap_{x \in T} \mathcal{E}_1(x)\Big).$$

Here $\mathcal{E}_1(x)$ captures the events that false detection was not raised, $\mathcal{E}_2(y)$ captures all the events when the algorithm raised a detection, and $\mathcal{E}_n$ captures the event that detection

was raised only around the region where the true changes occurred. The result holds if we establish two relations

On event $\mathcal{E}_n$, we have

$$\widehat{K} = K \ \& \ \max_{k \in \widehat{\mathcal{K}}} |\widehat{\tau}_k - \tau_k| \leq w, \ \text{and}$$

$$\mathbb{P}(\mathcal{E}_n^c) \to 0.$$

We begin by characterizing the probability of each event as separate results to highlight the assumptions required, and the main result follows from Lemmas 7-9.

**Lemma 7.** *Let* $\{\widetilde{X}_i\}_{i \in [n]}$ *be a collection that is input to* RC-Cat. *Let the threshold* $b = 2c_0\sqrt{\mathcal{M}\eta}$. *For* $x \in T$, *we have under assumption **A1***

$$\mathbb{P}(S_w(x) < b) \geq 1 - \delta.$$

*Proof.* As $x \in T$, by definition there is no change point in the interval $[x - w, x + w]$. Consider the random variables $\{\widetilde{X}_i\}_{i=x-w}^{x+w}$. Let $\mu_x$ denote the mean of the segment. Let $\Psi(\{\widetilde{X}_i\}_{i=x-w}^{x-1})$ and $\Psi(\{\widetilde{X}_i\}_{i=x+1}^{x+w})$ define the scan statistic in RC-Cat. By Corollary 2, we have using $w \geq \frac{B}{A\eta}\log(4/\delta)$,

$$|\Psi(\{\widetilde{X}_i\}_{i=x-w}^{x-1}) - \mu_x| \leq c_0\sqrt{\mathcal{M}\eta},$$
$$|\Psi(\{\widetilde{X}_i\}_{i=x+1}^{x+w}) - \mu_x| \leq c_0\sqrt{\mathcal{M}\eta},$$

each with probability at least $1 - \delta/2$. Therefore, the event $\mathcal{E}_1(x)$ occurs with probability at least $1 - \delta$. Indeed, by triangle inequality

$$S_w(x) = |\Psi(\{\widetilde{X}_i\}_{i=x-w}^{x-1}) - \Psi(\{\widetilde{X}_i\}_{i=x+1}^{x+w})| \leq 2c_0\sqrt{\mathcal{M}\eta}.$$

The result holds. $\square$

**Lemma 8.** *Let* $\{\widetilde{X}_i\}_{i \in [n]}$ *be a collection that is input to* RC-Cat. *Let the threshold* $b = 2c_0\sqrt{\mathcal{M}\eta}$. *Let assumptions **A1** - **A3** hold. For* $y \in \mathcal{K}$, *we have*

$$\mathbb{P}(S_w(y) > b) \geq 1 - \delta.$$

*Proof.* For any $k$, consider $y = \tau_k$. By assumption **A1** and **A2**, we have that the segment $\{\widetilde{X}_i\}_{i=\tau_k+1}^{\tau_k+w}$ has mean $\mu_{\tau_k}$ and the segment $\{\widetilde{X}_i\}_{i=\tau_k-w}^{\tau_k-1}$ has mean $\mu_{\tau_{k-1}}$. For simplicity, we abuse the notation and denote $\Psi(\{\widetilde{X}_i\}_{i=\tau_k+1}^{\tau_k+w}) := \Psi_1$, and $\Psi(\{\widetilde{X}_i\}_{i=\tau_k-w}^{\tau_k-1}) := \Psi_2$. We have using the inequality $(x + y)^2 \geq x^2/2 - y^2$ for any $x, y \in \mathbb{R}$,

$$\left|\left(\Psi_1 - \mu_{\tau_k}\right) - \left(\Psi_2 - \mu_{\tau_{k-1}}\right) + \left(\mu_{\tau_k} - \mu_{\tau_{k-1}}\right)\right|^2$$
$$\geq \theta^2/2 - \left|\left(\Psi_1 - \mu_{\tau_k}\right) - \left(\Psi_2 - \mu_{\tau_{k-1}}\right)\right|^2.$$

By Corollary 2, we have using $w \geq \frac{B}{A\eta}\log(4/\delta)$,

$$\left|\left(\Psi_1 - \mu_{\tau_k}\right)\right|^2 \leq b^2/4,$$
$$\left|\left(\Psi_2 - \mu_{\tau_{k-1}}\right)\right|^2 \leq b^2/4,$$

each with probability at least $1 - \delta/2$. The result follows using assumption **A3**. $\square$

**Lemma 9.** *Let the threshold* $b = 2c_0\sqrt{\mathcal{M}\eta}$. *Let assumptions **A1** - **A3** hold. On event* $\mathcal{E}_n$, *we have*

$$\{\widehat{K} = K\} \ \& \ \max_{k \in \widehat{\mathcal{K}}} |\widehat{\tau}_k - \tau_k| \leq w.$$

*Moreover,* $\mathbb{P}(\mathcal{E}_n^c) \to 0$.

*Proof.* First, note that $\widehat{\tau}_k \in T^c$, $\forall \ k \in \widehat{\mathcal{K}}$ by definition of $T$, where $\widehat{\tau}_k$ are the change points detected by RC-Cat. Therefore, we have that

$$\tau_k \in [\widehat{\tau}_k - w, \widehat{\tau}_k + w].$$

By assumption **A2**, we have that there are no other change points in this interval.

Next, we show that there is a change point identified in the interval $(\tau_k - w, \tau_k + w)$. Let $\lambda$ The intervals

$$\Omega_+ := (\tau_k + w, \tau_k + (\lambda+1)w) \ \& \ \Omega_- := (\tau_k - w, \tau_k - (\lambda+1)w)$$

are contained in $T$ by definition and **A2**. This implies that on every $x \in \Omega_+ \cup \Omega_-$, the event $\mathcal{E}_1(x)$ holds with the corresponding scan statistic $S_w(x) < b$. However, by Lemma 8 we have $S_w(\tau_k) > b$. This implies that there is a local maximum, say $\widehat{\tau}_k \in (\tau_k - w, \tau_k + w)$, and $S_w(\widehat{\tau}_k) \geq S_w(\tau_k) > b$.

Using union bound, we have

$$\mathbb{P}(\mathcal{E}_n^c) \leq \sum_{k \in \mathcal{K}} \mathbb{P}(\mathcal{E}_2^c(\tau_k)) + \sum_{x \in T} \mathbb{P}(\mathcal{E}_1^c(x)).$$

From Lemma 7 and Lemma 8, we have the trivial upper bound

$$\mathbb{P}(\mathcal{E}_n^c) \leq 2n\delta.$$

The result holds by choosing $\delta = \frac{1}{n^{c_1+1}}$ for $c_1 > 0$. $\square$

### 4.4 PROOF OF COROLLARY 5

Let $[m] := \{t : w + 1 \leq t \leq n - w\}$. As $\mathcal{K} = \emptyset$, we have that the event

$$\mathcal{E}_n = \cap_{x \in [m]} \mathcal{E}_1(x)$$

has a probability lower bound using Lemma 7 as

$$\mathbb{P}(\mathcal{E}_n) \geq 1 - n\delta,$$

where $\delta = \frac{1}{n^{c_1+1}}$ for $c_1 > 0$.

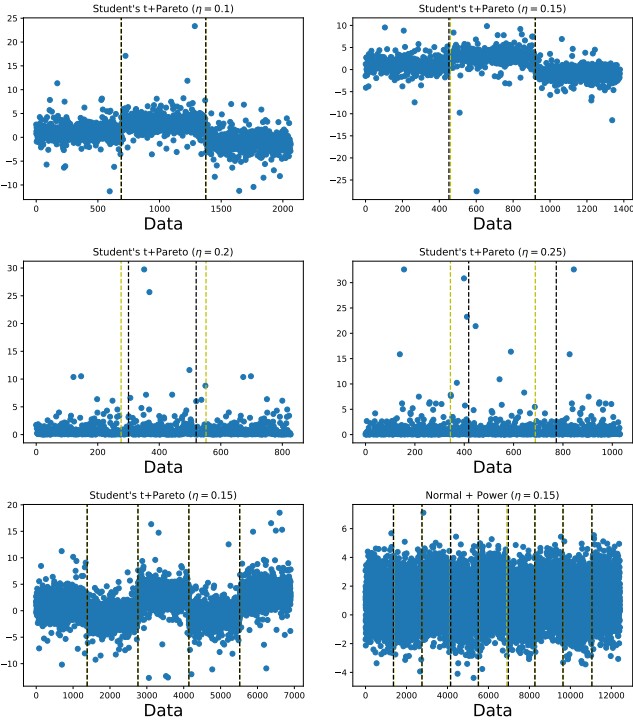

Figure 1: The figures show the performance of Algorithm 1 under different distributions for the outliers. The *yellow* line indicates the positions at which the distribution of the inliers changed, and the *black* dotted line shows the positions at which changes were announced by the algorithm. It can be seen that while the contamination level $\eta$ influences the offset – increases with increase in $\eta$ – it has limited bearing on the number of change points detected under a careful choice of the algorithm parameters.

## 4.5 PROOF OF THEOREM 3

The soft-truncation of $k$ samples as $\bar{\mu}_\eta$ obtains from Theorem 1, the deviation bound

$$|\bar{\mu}_\eta - \mu| \le \vartheta_k := \sqrt{2\mathcal{M}\Big(\frac{\log(4/\delta)}{k} + 2A\eta\Big)},$$

with probability at least $1 - \delta/2$. For the shifted data $\widetilde{X}'_i = \widetilde{X}_i - \bar{\mu}_\eta$ note that $\mathbb{E}[\widetilde{X}'^2|\mathcal{F}] \le (\mathcal{V} + \vartheta_k^2)$. So the soft-truncation estimate of shifted data obtains from Theorem 1, the deviation bound

$$|\mu'_\eta - (\mu - \bar{\mu}_\eta)| \le \sqrt{2(\mathcal{V} + \vartheta_k^2)\Big(\frac{\log(4/\delta)}{n - k} + 2A\eta\Big)},$$

with probability at least $1 - \delta/2$. Defining $\mu'_\eta := \widehat{\mu}_\eta - \bar{\mu}_\eta$, the result follows with probability $1 - \delta$ as both high-probability events should hold.

## 5 NUMERICAL RESULTS

In this section, we provide numerical results to illustrate the performance of Algorithm 1. Our main objective is to pro-

vide empirical evidence to support the consistency results.

## 5.1 SYNTHETIC DATA

We assume that the adversary/ nature replaces the original data with samples generated from random distributions. The algorithm parameters for all the figures in Figure 1 are chosen as follows. The inlier distributions are modeled as

$$X_t = \mu_t + \zeta_t,$$

where $\zeta_t$ is a martingale difference noise. From Theorem 3, the choice of $w$ is given as $w \ge \frac{B}{A\eta} \log(4/\delta)$ for a confidence level $\delta = 0.01$, $B = 2$, and $A = \log 2$. The value $\mu_t \le 3$, $\forall\, t$, obtaining a bound $\mathcal{M} = 10$ for unit variance. There is a trade-off between false detection and no-detection for different choices of $b$ informed by Theorem 1. For good performance, we recommend setting smaller than that informed by theory and increasing the neighbourhood width $\lambda$ for local search and elimination. For Figure 1, $\lambda = 3$ was chosen and $\alpha$ is informed by Theorem 1.

## 5.2 COMPARISON WITH ARC METHOD

In this section, we compare our method with a recent state-of-art method, the ARC algorithm [Li and Yu, 2021]. Specifically, we examine the robustness of proposed method and ARC under three different contamination settings.

(Setting 1) *Pareto contamination.* The inliers follow student-t distribution with degree of freedom 3. Outliers follow pareto distribution with degree freedom 2.

(Setting 2) *One-sided arbitrary contamination.* The inliers follow student-t distribution with degree of freedom 3. Outliers are fixed at 100.

(Setting 3) *Two-sided arbitrary contamination.* The inliers follow student-t distribution with degree of freedom 3. Outliers are fixed at 100 or -100.

The total time horizon $T$ is fixed at 1500, the confidence level $\delta = 0.01$, $A = \log(2), B = 2, \mathcal{M} = 5$, the true mean is in the range, $-3 \le \mu_t \le 3$ and *two* underlying change points equally spaced between $[0, T]$. We consider varying the following tuning parameter. The contamination rate $\eta \in \{5\%, 10\%, 20\%, 30\%, 40\%\}$. Window size $w \in \{80, 100, 120\}$. Average differences (i.e., average of $|\hat{\tau}_k - \tau_k|$'s) between detected time and true change points are reported. To be fair (without deliberately tuning threshold $b$), for both methods, the detected change points are chosen to be time stamps with top two $S_w(k)$ values.

Based on the Tables 1 - 3, we can find that the proposed method is robust to different contamination level, while ARC method is not. Especially when we increase contamination rate $\eta$ to 40 %, ARC behaves much worse. Moreover, our method is also less sensitive to the choices of window

| Setting 1 | | | | | |
|---|---|---|---|---|---|
| $w = 80$ | | | | | |
| $\eta$ | 0.05 | 0.1 | 0.2 | 0.3 | 0.4 |
| Ours | 6.5 | 13.3 | 31.3 | 43.3 | 55.9 |
| ARC | 18.5 | 26.1 | 40.9 | 44.2 | 58.6 |
| $w = 100$ | | | | | |
| $\eta$ | 0.05 | 0.1 | 0.2 | 0.3 | 0.4 |
| Ours | 3.6 | 6.6 | 14.8 | 26.4 | 31.2 |
| ARC | 19.8 | 20.2 | 22.6 | 32.7 | 36.9 |
| $w = 120$ | | | | | |
| $\eta$ | 0.05 | 0.1 | 0.2 | 0.3 | 0.4 |
| Ours | 2.9 | 3.4 | 7.0 | 7.7 | 17.7 |
| ARC | 17.7 | 18.0 | 19.9 | 23.5 | 25.6 |

Table 1: The table of detection error under Setting 1 with various choices of tuning parameters $\eta$ and $w$. Each case is replicated for 500 times.

| Setting 2 | | | | | |
|---|---|---|---|---|---|
| $w = 80$ | | | | | |
| $\eta$ | 0.05 | 0.1 | 0.2 | 0.3 | 0.4 |
| Ours | 2.0 | 3.5 | 9.6 | 25.8 | 43.5 |
| ARC | 15.3 | 14.4 | 19.7 | 134.6 | 46.2 |
| $w = 100$ | | | | | |
| $\eta$ | 0.05 | 0.1 | 0.2 | 0.3 | 0.4 |
| Ours | 2.0 | 2.9 | 13.4 | 21.3 | 44.9 |
| ARC | 13.8 | 14.5 | 34.5 | 119.4 | 98.7 |
| $w = 120$ | | | | | |
| $\eta$ | 0.05 | 0.1 | 0.2 | 0.3 | 0.4 |
| Ours | 2.1 | 2.5 | 11.8 | 25.6 | 44.0 |
| ARC | 16.0 | 14.5 | 20.4 | 81.0 | 84.7 |

Table 2: The table of detection error under Setting 2 with various choices of tuning parameters $\eta$ and $w$.

| Setting 3 | | | | | |
|---|---|---|---|---|---|
| $w = 80$ | | | | | |
| | 0.05 | 0.1 | 0.2 | 0.3 | 0.4 |
| Ours | 2.6 | 3.9 | 10.6 | 25.5 | 36.5 |
| ARC | 14.8 | 14.3 | 13.9 | 81.8 | 130.7 |
| $w = 100$ | | | | | |
| | 0.05 | 0.1 | 0.2 | 0.3 | 0.4 |
| Ours | 2.9 | 3.7 | 10.0 | 21.0 | 34.1 |
| ARC | 15.0 | 13.7 | 18.8 | 97.1 | 96.1 |
| $w = 120$ | | | | | |
| | 0.05 | 0.1 | 0.2 | 0.3 | 0.4 |
| Ours | 2.6 | 3.6 | 9.1 | 15.2 | 32.2 |
| ARC | 15.7 | 14.7 | 14.0 | 70.6 | 62.2 |

Table 3: The table of detection error under Setting 3 with various choices of tuning parameters $\eta$ and $w$.

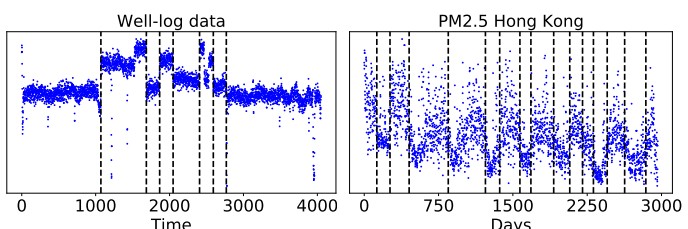

Figure 2: The detection results returned by `RC-Cat` for two real-world data sets, well-log data and PM2.5 index data. In the experiment, we choose tuning parameters $b = c_0\sqrt{\mathcal{M}\eta}/2$ and $\lambda = 1$.

From Figure 2, we can see that the proposed method `RC-Cat` can well detect the jump points in well-log data and is very robust to those outliers. Our method can also capture the fluctuations of Hong Kong PM2.5 index.

size than ARC method. These results indicate that `RC-Cat` is indeed a better method.

## 5.3 REAL-WORLD DATA

We consider two real data sets in this subsection, the *well-log* data [Jeremias, 2018, Fearnhead and Rigaill, 2019, Li and Yu, 2021] which has been widely studied in the existing literature and *PM2.5 index* data [URL, 2018] which has not been considered in the literature.

**Well-log** data set contains 4050 measurements of nuclear magnetic response during the drilling of a well. Majority of the observations behave very well and a small proportion of the observations are far away from the mean value.

**PM2.5 index** data set records air quality of Hong Kong during 1-Jan 2014 to 2-Feb-2022. The PM2.5 index fluctuates occasionally over the total period of time.

## 6 CONCLUSION

In this work, we have provided a robust change detection algorithm based on influence functions that can deal with a fraction of arbitrary but weakly adversarial contamination. Key contributions to the vast literature on robust offline change detection methods include: (i) The ability to handle non-i.i.d data along with contamination, when minimal assumptions are made on the distributions of the inliers. (ii) A computationally appealing algorithm that is consistent. The algorithm itself is intuitive, and combines local search methods to segment the dataset. Also, empirical results confirm that the algorithm outperforms the state of the art offline change detection algorithm in terms of average detection times, demonstrating significant gains under heavy-contamination. This work motivates change detection in multi-variate datasets, possibly in the presence of contamination, motivated by the appealing aspect of obtaining dimension-free robust estimation in high-dimension using influence functions; see Catoni and Giulini [2017].

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
