# OpenReview forum: "Offline Change Detection under Contamination"
_auai.org/UAI/2022/Conference — UAI 2022 Poster_

### Official Review · Reviewer_zC4u · 2022-04-06

**Q2(1) Originality/Novelty:** 2
**Q2(2) Significance/Impact:** 2
**Q2(3) Correctness/Technical Quality:** 3
**Q2(6) Clarity Of Writing:** 3
**Q6 Overall Score:** 6
**Q8 Confidence In Your Score:** 4

**Q1 Summary And Contributions:**

The authors develop an algorithm for offline change detection, i.e. they describe a method for the statistical estimation of the time points at which there is a change in the generating distribution of a sequence of samples. The algorithm leverages robust estimators of the mean(s) of the generating distribution(s), based on influence functions. The authors demonstrate that their method is consistent, and illustrate the result through a number of numerical experiments.

**Q2 Assessment Of The Paper:**

More detailed information regarding each of these aspects is given below:

**Q2(4) Quality Of Experiments (Optional):**

2: Fair: The experimental evaluation is weak: important baselines are missing, or the results do not adequately support the main claims.

**Q2(5) Reproducibility:**

3: Good: Key resources (e.g., proofs, code, data) are available and key details (e.g., proofs, experimental setup) are sufficiently well-described for competent researchers to confidently reproduce the main results.

**Q3 Main Strengths:**

The authors clearly explain their use of the non-parametric robust mean estimators based on influence functions. The incorporation of these estimators in the change point detection algorithm is clear and intuitive, and the authors give a thorough argument demonstrating the consistency of their method. The experimental analysis gives a fair indication of the type of results that can be obtained by the algorithm.

**Q4 Main Weakness:**

The experimental results section seems somewhat short; I would have liked to see some more details regarding the different choice of parameters, more details about the performance of the method in different settings, and more (explicit) comparisons with related methods from the literature. In particular the lack of such comparisons makes it difficult to assess the relative merit of the proposed method.

**Q5 Detailed Comments To The Authors:**

Overall I think the paper is well-structured and clearly written. Throughout, there appear minor (but frequent) grammatical mistakes, in particular missing articles and incorrect pluralization. If the paper is accepted, I would be really interested in seeing the experiments section expanded. This might be a good use of the additional page length allowed for the final version. Alternatively, I think that some of the proofs from Section 4 might be provided as supplementary material, which would also free up some space.


**Q7 Justification For Your Score:**

The paper appears to be technically sound and fits in the scope of the UAI conference. However, based on the content of the paper, I find it difficult to judge the impact or merits of the method proposed in this work.

**Q9 Complying With Reviewing Instructions:**

1: Yes.

---

### Official Review · Reviewer_hxpT · 2022-04-12

**Q2(1) Originality/Novelty:** 3
**Q2(2) Significance/Impact:** 3
**Q2(3) Correctness/Technical Quality:** 3
**Q2(6) Clarity Of Writing:** 4
**Q6 Overall Score:** 5
**Q8 Confidence In Your Score:** 3

**Q1 Summary And Contributions:**

The authors provide a novel offline change detection algorithm that is robust to a specific model of weakly adversarial input. Their algorithm is designed to work on real world (i.e. non-iid) data and takes the form of a scan statistic. This improves on previous work within the scan statistic and change detection literature by it's focus on non-iid data and adversarial noise models.

**Q2 Assessment Of The Paper:**

More detailed information regarding each of these aspects is given below:

**Q2(4) Quality Of Experiments (Optional):**

2: Fair: The experimental evaluation is weak: important baselines are missing, or the results do not adequately support the main claims.

**Q2(5) Reproducibility:**

3: Good: Key resources (e.g., proofs, code, data) are available and key details (e.g., proofs, experimental setup) are sufficiently well-described for competent researchers to confidently reproduce the main results.

**Q3 Main Strengths:**

- Great writing, very easy to follow the paper's development. The literature was very helpful for grounding the work.
- Great building on existing methodology while pushing for more modern concerns such as adversarial contamination.
- Theoretical results combined with empirical analysis provides a great resource for understanding how the algorithm operates

**Q4 Main Weakness:**

- The motivation in terms of application was not totally clear. While the features of the algorithm are all great (non-iid, adverserially robust, fast, etc.) I'm not sure of the use case(s) that the authors had in mind.
- The empirical results section could be substantially improved. In my mind that is the weakest part of the paper
-- The results are presented only in a number of scatter plots, no summary stats/charts
-- No comparison algorithms are used
-- No analysis of what was missed, for example as adversarial contamination increased

**Q5 Detailed Comments To The Authors:**

See the comments above. Really enjoyed reading the paper and it seems like a very solid basis from which to grow. Two specific questions:

- It's unclear to me how generalizable this adversarial model is. Meaning, if a slightly different (but still "weak" adversarial model was used, would the method and results fall apart?
- Regarding the worst case speed, could you incorporate an Linear Time Subset Scanning approach to provide greater efficiency for worst case of nlon(n)? Or is that not applicable in your context?

**Q7 Justification For Your Score:**

- The method is novel, though not groundbreaking.
- The theoretical results are compelling
- If the empirical section were stronger I would easily upgrade the score to 6-7

**Q9 Complying With Reviewing Instructions:**

1: Yes.

---

### Official Review · Reviewer_kWvz · 2022-04-17

**Q2(1) Originality/Novelty:** 2
**Q2(2) Significance/Impact:** 2
**Q2(3) Correctness/Technical Quality:** 3
**Q2(6) Clarity Of Writing:** 3
**Q6 Overall Score:** 5
**Q8 Confidence In Your Score:** 3

**Q1 Summary And Contributions:**

This paper proposed a non-parametric change detection algorithm to detect multiple change points in time series data under non-adversarial contamination. The algorithm is designed for the offline setting, where the objective is to detect changes when all data are received. This method has the ability to handle the non-i.i.d data along with contamination, where the inliers only have a bounded second moment, and provide a computationally appealing algorithm that is consistent.

**Q2 Assessment Of The Paper:**

More detailed information regarding each of these aspects is given below:

**Q2(4) Quality Of Experiments (Optional):**

2: Fair: The experimental evaluation is weak: important baselines are missing, or the results do not adequately support the main claims.

**Q2(5) Reproducibility:**

3: Good: Key resources (e.g., proofs, code, data) are available and key details (e.g., proofs, experimental setup) are sufficiently well-described for competent researchers to confidently reproduce the main results.

**Q3 Main Strengths:**

1. The motivation of this paper is to deal with the inherent challenges in change point detection scenarios such as dealing with non-i.i.d data and outliers, dealing with the introduction of adversarial examples in dataset. The motivation of this paper is clear and well stated.

2. The structure of this paper is well organized.

3. The technique part of this paper seems to be sound, although it lacks novelty to some extent. The majority of the techniques used in this paper are a combination of existing results.  Section 3 and Section 4 provide plenty of theorems, detailed proofs, and mathematical formula derivations to construct the proposed algorithms. Although I have not verified the accuracy of all the proofs, these proofs and derivations lay the foundation for the reliability and trustworthiness of the proposed algorithm.

**Q4 Main Weakness:**

1. There are several inappropriate words and phrases in this paper, resulting in some sentences not smooth, making this paper less readable. See below.
2. Although related work has been introduced in Section 1.2, there are still many citations in the latter part of this paper, causing some inconvenience for the readers. Moreover, the format of citations in this paper is not consistent, sometimes they are “author name [published year]”, sometimes they are “[author name, published year]”
3.The proofs of the theorems are not in order.


**Q5 Detailed Comments To The Authors:**

Questions:
1. It is helpful to explain the abbreviation “r.vs” for readers without enough background knowledge.

2. I know that A1-A3 represent three assumptions. But it is better to explain the meaning of the notation C1-C2 to ensure that the overall notations and their explanations (You can name C1-C2 as conditional expected values C1-C2 and name A1-A3 as assumptions A1-A3) of this paper are consistent.

3. The readers may need a more detailed and clear explanation of the experiment results, explaining what we can conclude from these figures and experiment results, especially for the experiments on synthetic data.

4. The accuracy of the algorithm in detecting multiple change points in time series data has not been experimentally verified. The results of experiments on real-world data are great, the proposed method RC-Cat can well detect the jump points in well-log data and can also capture the fluctuations of Hongkong PM2.5 index. But the accuracy of the algorithm is not very good under some settings of the experiments on synthetic data, such as “Student’s t + Pareto () and Student’s t + Pareto ()”. It is recommended to select some baseline models and do experiments to verify the accuracy of the proposed algorithm.

5.In experiment on synthetic data with the setting “Normal + power ()”, the performance of the proposed algorithm is quite well. The positions announced by the algorithm is quite close to the true position. But these yellow lines and black lines overlapped so that we even need to zoom in this figure to check the experiment results. Personally, I suggest that the authors try to make this figure more distinguishable.


Typos:
1.In Point 1 of Section 1.1, there is an unnecessary extra “the” in the sentence “This additional sophistication of local search methods was the introduced in Niu and Zhang…”.
2. First paragraph in Section 1.2, “Change detection methods are mainly classified studied in online and offline settings”, this sentence does not read smoothly.
3. First paragraph in Section 2.1, use “corruption rates” instead of “corruption rate”.
4. First paragraph in Section 2.2, “The idea of using (convex) influence functions for robust mean estimation is not new Huber[2004]”, this sentence does not read smoothly. Maybe “The idea of using (convex) influence functions for robust mean estimation is not new. [Huber, 2004]” is better.
5.  In Section 5.2, it should be “PM 2.5” instead of “PM25”.
6. First paragraph of Section 6, it should be “The robustness feature of the algorithm was introduced…” instead of “The robustness feature of the algorithm was introduce…”.
7. The experiments only considered simple and toy cases. More challenging examples may be considered.

**Q7 Justification For Your Score:**

The structure of this paper is well organized.

The technique part of this paper seems to be sound.

**Q9 Complying With Reviewing Instructions:**

1: Yes.

---

### Decision · Program_Chairs · 2022-05-15

**Decision:**

Accept (Poster)

**Comment:**

Meta Review: The paper presents a novel approach to offline change detection in time series data. The approach is supported by consistency results and empirical evaluation. The AC and reviewers all agree that the paper makes some interesting contributions.  The author feedback to the reviews greatly improves the significance of this work and we therefore urge the authors to  incorporate their points and the additional experimental results in their revision.